# Overcoming Acquired Drug Resistance to Cancer Therapies through Targeted STAT3 Inhibition

**DOI:** 10.3390/ijms24054722

**Published:** 2023-03-01

**Authors:** Sunanda Singh, Hector J. Gomez, Shreya Thakkar, Samara P. Singh, Ashutosh S. Parihar

**Affiliations:** 1Singh Biotechnology, 1547 Fox Grape Loop, Lutz, FL 33558, USA; 2Department of Cell Biology, Microbiology and Molecular Biology, University of South Florida, Tampa, FL 33620, USA; 3Department of Surgery, Division of Surgical Oncology, University of Miami Miller School of Medicine, Miami, FL 33136, USA

**Keywords:** STAT3, acquired drug resistance, kinase inhibitors, chemotherapy, monoclonal antibodies, immune checkpoint inhibition

## Abstract

Anti-neoplastic agents for cancer treatment utilize many different mechanisms of action and, when combined, can result in potent inhibition of cancer growth. Combination therapies can result in long-term, durable remission or even cure; however, too many times, these anti-neoplastic agents lose their efficacy due to the development of acquired drug resistance (ADR). In this review, we evaluate the scientific and medical literature that elucidate STAT3-mediated mechanisms of resistance to cancer therapeutics. Herein, we have found that at least 24 different anti-neoplastic agents—standard toxic chemotherapeutic agents, targeted kinase inhibitors, anti-hormonal agents, and monoclonal antibodies—that utilize the STAT3 signaling pathway as one mechanism of developing therapeutic resistance. Targeting STAT3, in combination with existing anti-neoplastic agents, may prove to be a successful therapeutic strategy to either prevent or even overcome ADR to standard and novel cancer therapies.

## 1. Introduction

Cancer encompasses a diverse group of diseases with common features and behaviors [1]. Within each histologic type of malignancy, there is often tremendous heterogeneity, which develops from the highly unstable genome of cancer cells. This genomic instability leads to development of cancer cell variants within the bulk tumor. When selection pressure is applied by exposing cancers to anti-neoplastic treatments such as chemotherapeutics, targeted therapeutics, anti-hormone therapeutics, and monoclonal antibodies, clonal selection may occur in patients, and oftentimes acquired drug resistance (ADR) develops. 

Drug resistance is a very complex and heterogenous problem that has developed through a variety of many mechanisms. Even within the same patient, several modes of ADR may be present across different tumors [2]. Drug resistance can be intrinsic (i.e., de novo) or conditional to an initial response to an anti-neoplastic agent followed by progression of the cancer—otherwise known as ADR. Signal transducer and activators of transcription 3 (STAT3) has been implicated in the development and maintenance of ADR in multiple cancers in response to various therapies. In this review, we will focus on the role of STAT3 in the development of ADR and clinically relevant drugs that are susceptible to STAT3-mediated ADR. Combination therapy of STAT3 inhibitors with therapies prone to ADR may prove to be synergistic and a compelling strategy to overcome therapeutic resistance in the clinical setting. 

Human cells have evolved to develop complex regulatory mechanisms, including positive feedback loops and significant crosstalk among oncogenic signaling pathways. In its simplistic form, ADR can develop when the inhibition of one pathway induces the activation of another, which may impair any therapeutic effect. Twelve pathways, critical to ADR, have been identified, and the STAT3 signaling pathway, described as the “master regulator of antitumor immune response” is one of them [3,4]. 

In general, the anti-neoplastic agents that have limited efficacy as result of the development of ADR can be divided into four groups: (1) traditional chemotherapeutic drugs, (2) targeted therapeutics, (3) anti-hormone therapeutics, and (4) monoclonal antibodies (Table 1) [5,6,7,8,9,10,11,12,13,14,15,16,17,18,19,20,21,22,23,24,25,26,27,28,29,30]. The traditional chemotherapeutic drugs that develop ADR and are discussed here include doxorubicin [31,32,33,34], gemcitabine [35,36,37,38,39,40,41,42,43,44], cisplatin [34,45,46,47,48,49,50,51,52,53,54,55], temozolomide [56,57,58,59,60], and paclitaxel [61,62,63,64,65]. Targeted therapeutics for which the development of ADR has been documented included in this review are afatinib [37,60,66,67], crizotinib [68], dasatinib [69], and erlotinib [70,71,72,73,74]. The anti-hormone therapeutics that develop ADR are flutamide [75,76,77], enzalutamide [78,79], and tamoxifen [80,81,82,83]. ADR is also developed to monoclonal antibodies such as cetuximab [84,85,86], bevacizumab [87], trastuzumab [88,89,90,91], and to the immune checkpoint inhibitors (ICIs) [92,93,94,95,96,97,98,99,100,101,102,103,104,105,106], such as pembrolizumab [101], nivolumab [102,104], and ipilimumab [103]. In response to all these anti-neoplastic agents, cancer cells utilize STAT3 as one mechanism of escaping their therapeutic effects and promoting ADR. There is a very large body of scientific and medical literature to support the use of anti-STAT3 therapeutics to overcome ADR in these cases. While other mechanisms of ADR exist, here we focus on STAT3 as a key mechanism for the development of ADR.

It should be stated that STAT3 is present in all mammalian cells and plays an important role in physiological functions. Under normal conditions the duration of STAT3 activity is short and transient but in pathological situations, such as cancer, a stronger activation is maintained over long periods of time [117,118]. Activated STAT3 refers to the phosphorylated STAT3 (tyrosine or serine phosphorylated) and is measured and or quantified and described as p-STAT3. This pathological form is referred to in the literature by many different terms such as aberrant, constitutive, dysregulated, etc. STAT3 present in cancer cells is p-STAT3, the constitutively phosphorylated form responsible for the acquired resistance described in this publication [117,118]. STAT3 is located intracellularly, downstream many kinases at an exchanging point of the most important signaling pathways involved in cancer. Oftentimes, when an administered drug blocks a specific kinase pathway, the STAT3 pathway is triggered as result of the crosstalk amongst upstream pathways, resulting in the aberrantly persistent form of p-STAT3.

## 2. STATs (Signal Transducer and Activators of Transcription)

There are seven STATs (STAT1, STAT2, STAT3, STAT4, STAT5a, STAT5b, and STAT6) that are intracellular proteins which function as signal messengers and transcription factors. They transmit signals from cytokines, growth factors, intracellular kinases, mutated oncoproteins, and other signaling pathways to the nucleus. Tyrosine phosphorylation cascade occurs after ligand binding by many extracellular molecules such as epidermal growth factor (EGF), platelet-derived growth factor (PDGF), fibroblast growth factor (FGF), interleukin-6 (IL-6), IL-5, oncostatin-M (OSM), granulocyte colony stimulating factor (GCSF), colony stimulating factor-1 receptor (CSF1R), leukemia inhibitory factor (LIF), c-kit, c-Met, insulin receptor, angiotensin-II receptor (AgtR2), interferons (IFNs), G-protein coupled receptors (GPCRs), and others. After such ligands bind the extracellular portion of their receptors, their intracellular portion attracts the Janus Kinase family (JAK1, JAK2, JAK3, and Tyk2) of proteins, which become phosphorylated. The JAK protein then phosphorylates STAT3 (pSTAT3) at tyrosine 705 and sometimes serine 727 to activate STAT3. Other intracellular kinases, which can directly activate STAT3 are Src and BCR–ABL, the mutant fusion protein in chronic myelogenous leukemia (CML) [26]. P-STAT3 then forms dimers, which translocate to the nucleus via chaperone proteins. There p-STAT3 dimers bind to specific nine base pair sequences in regulatory genomic regions to regulate transcription of specific genes. The signaling function of p-STAT3 is carefully regulated by inhibitory molecules such as protein inhibitors of activated STAT (PIAS), protein tyrosine phosphatases (PTPases), and suppressors of cytokine signaling (SOCS). Dysregulation of the normal physiologic balance of p-STAT3 and unphosphorylated STAT3 can occur due to upstream mutations or protein overexpression. This results in constitutive expression of p-STAT3 and continuous transcription of pro-oncogenic and anti-apoptotic genes, which promotes cancer growth, proliferation, cell cycle re-entry, angiogenesis, immunosuppression, and, metastasis when anticancer agents apply selective pressure might induce the development of ADR.

## 3. The Role of STAT3 in Cell Cycle Arrest and Regulation

STAT3 play critical roles within neoplastic cells, immune cells, and other stromal cells, such as cancer-associated fibroblasts (CAFs). Once activated within tumor cells, phosphorylated STAT3 (p-STAT3) regulates the transcription of various immunosuppressive cytokines such as VEGF, IL-10, and TGF-β. p-STAT3 can promote tumor progression by increasing transcription of genes associated with stemness and epithelial to mesenchymal transition (EMT) [119]. Additionally, p-STAT3 is involved in two apoptotic processes, autophagy and anoikis, both contributors to ADR development.

Autophagy, a cellular degradation process, is another regulatory mechanism that plays a major role in maintaining homeostasis, and its dysfunction has been implicated in cancer progression and ADR. The signaling pathways that control inducible autophagy and cell death are closely associated and incorporated into the tumor regulatory network of autophagy related proteins, ultimately affecting the fate of tumor cells [120]. The crosstalk between autophagy and other stress response pathways including STAT3, determines the survival or death of a cell. Nuclear STAT3 regulates autophagy in various forms. For instance, STAT3 inhibits autophagy by activating BCL2 or increases it by upregulating and stabilizing HIF1A under hypoxia; however, it has been determined that cytoplasmatic STAT3 regulates autophagy in a more direct way [121]. Autophagy initially prevents cancer progression but under stressful situations improves the survival of cancer cells [122] contributing to ADR and therapy failure. p-STAT3 has been found to be associated with aberrant autophagy activity in many oncological studies [123]. The anti-autophagy action is partly due to STAT3-mediated inhibition of the BEBN1/PIKC3 complex, resulting in reduced Beclin-1 activity. There is a link between ADR to chemotherapeutics, sometimes described as chemoresistance, and autophagy. The autophagic process vary depending on the tumor stage. In some cases, high dosage chemotherapy may induce protective autophagy that leads to ADR. Some proteins such as mTOR, Beclin-1, miRNA, and autophagy-related genes play a role during treatment of some cancers such as osteosarcoma. The use of autophagy inhibitors in combination with chemotherapeutics is being studied as a new treatment of cancer that might avoid chemoresistance [124]. STAT3 inhibition increases autophagy by increasing transcription of key activators of autophagy. [125]. The importance of autophagy in tumor immunity and ADR is now recognized and has been reported that optimal induction or inhibition of autophagy may induce effective treatments when combined with immunotherapy [126].

Anoikis, another type of apoptosis, is triggered by loss of cell adhesion [127]. It might be activated during tumorigenesis to clear off extracellular matrix (ECM) and detached cells. Cancer cells that develop the ability to survive are called anoikis-resistant cells. These cells become very aggressive and drug resistant, developing the capacity to invade and migrate to metastatic sites. Several features have been identified as responsible for modulating anoikis resistance, one of which is STAT3. STAT3-related anoikis-resistance has been reported in cancer cells of human pancreatic cancer, melanoma, cholangiocarinoma, esophageal squamous cell carcinoma, squamous cell carcinoma, nasopharyngeal carcinoma, and lung carcinoma [128,129,130,131,132]. These cancer cells were reported to have enhanced cell migration, invasion capability and high metastatic potential, and inhibition of STAT3 led to sensitization of all those anoikis-resistant cells [133].

Nicotinamide N-methyltransferase (NNMT) participates in the development of ADR. NNMT, a cytoplasmic enzyme that methylates nicotinamide, is regulated by STAT3 and has been shown to be overexpressed in solid tumors. Furthermore, STAT3 activation intensifies the expression of NNMT and stimulates its activity [134]. NNMT has been identified as an oncogene in intrahepatic cholangiocarcinoma [135]. NNMT is upregulated in cutaneous squamous cell carcinoma, induces cellular invasion via EMT-related gene expression [136] and plays critical roles in the incidence and development of various cancers [137].

Evidence that NNMT plays an important role in cancer can be seen by the fact that NNMT knockdown reduces tumorigenesis and chemoresistance and that Yuanhuadine, a natural inhibitor of NNM, reverses EGFR inhibitors ADR [138]. Chemoresistance or ADR to adriamycin and paclitaxel in breast cancer has been also reported by Wang et al., 2019. This group found that reversal of NNMT related ADR can be accomplished by using the SIRT1 inhibitor, EX527 or using siRNA therapy. SIRT1 also represses the activation of STAT3 and NF-κB proteins via deacetylation [139].

The major function of the tumor suppressor p53 is to induce transient cell cycle arrest, cellular senescence, and apoptosis, a significant barrier to the development of tumors; however, p-STAT3 can inhibit these repressive functions of p53 [140]. This crosstalk between STAT3 and p53 also contributes to the development of ADR and the loss of the pharmacologic effects of anticancer agents [141]. STAT3 inhibition upregulates the expression of p53 and increases cellular apoptotic activity, thereby reversing ADR. Another important signaling pathway for growth and proliferation is the RAS/mitogen activated pathway kinase (MAPK). The crosstalk between STAT3 and p53/RAS signaling regulates metastasis and cisplatin resistance in ovarian cancer through the Slug/MAPK/PI3K/AKT-mediated regulation of EMT and autophagy [142]. Therefore, RAS and STAT3 activation promote ovarian cancer growth, metastasis, and cisplatin resistance. Dual inhibition of STAT3 and KRAS, achieved by nano-antibody SBT-100, would be an ideal treatment for this type of cancer to overcome ADR in ovarian and many other types of cancer [143].

As previously mentioned, p-STAT3-mediated ADR occurs in response to anti-neoplastic agent therapy by utilizing multiple intracellular signaling pathways. As illustrated in Figure 1, treatment with a receptor tyrosine kinase inhibitor, which blocks MAPK pathways, results in the cancer cells secreting ligands, which bind to receptors on the cancer cells themselves in an autocrine fashion or to CAFs, intratumor macrophages, and other cells in the tumor microenvironment (TME) in a paracrine fashion. This ligand binding to its cognate receptor results in STAT3 activation, turning on numerous genes that promote growth, proliferation, cell cycle re-entry, anti-apoptosis, angiogenesis, immunosuppression, and metastasis, and ultimately circumventing the anti-neoplastic therapy being used resulting in ADR.

## 4. ADR Development to Chemotherapeutics

Doxorubicin is an anthracycline compound and currently one of the most effective classes of anti-cancer agents in clinical applications; however its use is limited by its chronic and acute toxicities [107]. It binds to topoisomerase I and II, resulting in intercalation of the base pairs of the DNA double helix and inhibition DNA replication. Because of this mechanism of action, doxorubicin has been highly effective in treating a wide variety of malignancies. Its efficacy is unfortunately limited in many cases by ADR due to STAT3 upregulation. A well-known STAT3 inhibitor, Stattic, was formulated in nanostructured lipid carriers to enhance the efficacy of doxorubicin against melanoma cancer cells [108,109]. Doxorubicin induces p-STAT3 in human breast cancer MCF cell line (ER+, non-metastatic) and human triple negative breast cancer MDA-MB-231 cell line (metastatic) [110]. The p-STAT3 was then suppressed by tyrphostin AG490 (an inhibitor of the upstream activating Janus kinases), transfection with a dominant-negative form of STAT3, and with satraplatin (a tetravalent platinum derivate that inhibits STAT3 phosphorylation) [110]. These treatments downregulated p-STAT3 and sensitized the cancer cells to doxorubicin.

Alantolactone (ALT), a sesquiterpene lactone component of Inula helenium, has anti-neoplastic effect against a variety of malignancies. Mechanistic research demonstrated that ALT abrogated STAT3 phosphorylation by promoting STAT3 glutathionylation. Reactive oxygen species scavenger NAC reverted ALT-mediated STAT3 glutathionylation and abrogation of STAT3 activation. With lung adenocarcinoma (A549 cell line), STAT3 inhibition by ALT enhanced chemosensitivity to doxorubicin via oxidative stress [111]. In the above three examples, genes transcribed by p-STAT3 dimers that are necessary for malignant cell behavior include *BCL2L1* (Bcl-xL), *BIRC5* (survivin), *HIF1A, HIF1B* (HIF-1), and *MMP9* had their expression reduced [145].

Human osteosarcoma (SJSA-1) tumors when treated in vivo with doxorubicin undergoes significant growth suppression during a 14-day course of treatment; however, only 28% of the treated mice survived the 3-week xenograft study. The doxorubicin-associated toxicity was killing the mice. It is not clear if doxorubicin’s effect on normal cells caused this, or the induction of p-STAT3 in the SJSA-1 cells or a combination of both may have contributed to the death of the mice. When xenograft mice received doxorubicin with SBT-100, a STAT3 inhibitor, the osteosarcoma tumor growth was significantly suppressed, and survival of the mice increased to 71%. By some mechanism, SBT-100 was protecting the mice from doxorubicin toxicity. SBT-100 is a camelid derived single domain antibody that penetrates the cell membrane and blood brain barrier (BBB) rapidly in vivo and inhibits STAT3. SBT-100 has broad range of efficacy against many human malignancies such as ER + PR+ breast cancer, HER2-amplified breast cancer, triple negative breast cancer (TNBC), pancreatic cancer, prostate cancer, glioblastoma, osteosarcoma, fibrosarcoma, and leukemia [143].

Cisplatin is a platinum-based anti-neoplastic agent that binds DNA and inhibits its replication. It is used to treat ovarian, cervical, testicular, head and neck, colorectal, esophageal, bladder, lung, and breast cancers. Some mechanisms by which cisplatin resistance can develop include decreased drug import, increased drug export, increased DNA damage repair, increased drug inactivation by detoxification enzymes, and inactivation of cell death signaling, which occur within cancer cells [146]. Another mechanism of cisplatin resistance involves STAT3 overexpression. Sun et al have summarized utilization of STAT3 inhibition to reverse cisplatin induced resistance [52]. They summarized a large variety of STAT3 inhibitors which reverse cisplatin resistance in lung cancer, ovarian cancer, cervical cancer, breast cancer, laryngeal cancer, head and neck cancers, esophageal cancer, and hepatocellular carcinoma [52].

Morelli et al found, through network analysis and classification of proteome analysis of A549 cells (lung adenocarcinoma), that there were pathways altered in cisplatin resistant A549 cells. The resistance profile of these A549 cells included STAT3 overexpression. Furthermore, p-STAT3 is a marker of poor prognosis and cisplatin resistance in lung cancer. Generation of A549 STAT3 knockout cells resulted in impairment of clonogenic survival and mesenchymal phenotype in these A549 cells. These STAT3 knockouts do not develop cisplatin resistance nor over activation of mammalian target of rapamycin (mTOR) signaling with cis treatment. Moreover, the A549 knockout cells are more sensitive to mTOR inhibition by rapamycin [147].

Ovarian cancer can be effectively treated with paclitaxel; however, the development of resistance remains a major problem. Sheng et al have shown that STAT3 directly activates the pentose–phosphate pathway to cause pro-oncogenic behavior of paclitaxel resistant ovarian cancer [112]. Furthermore, they discovered that STAT3, p-STAT3, and glucose-6-phosphate dehydrogenase (G6PD) protein levels are increased in paclitaxel resistant cell lines versus paclitaxel sensitive cell lines. Blocking STAT3 decreased *G6PD* mRNA expression and increased the sensitivity of paclitaxel resistant ovarian cancer cells to paclitaxel. In addition, they demonstrated that STAT3 directly binds to the *G6PD* DNA promoter region and increases the expression of *G6PD* at the transcriptional level. In summary, their research reveals overexpression of STAT3 increases ovarian cancer colony formation, proliferation, and resistance to paclitaxel by increasing *G6PD* expression and pentose–phosphate metabolism flux [112].

With cervical cancer, paclitaxel is an important chemotherapeutic agent, but here too resistance to paclitaxel develops. Fan et al compared microRNA (miRNA) expression in cervical cancer cell lines to their paclitaxel resistant cervical cancer counterparts [113]. They found *miR-125a* to be a significantly decreased miRNA among paclitaxel-resistant cervical cancer cells and these cells also developed cisplatin resistance. Upregulating miR-125a sensitized resistant cervical cancers to paclitaxel in vitro and in vivo and to cisplatin in vitro. Importantly, they showed *miR-125a* increased sensitivity of cervical cancers to paclitaxel and cisplatin by decreasing STAT3. *MiR-125a* improved paclitaxel and cisplatin sensitivity by causing chemotherapy induced apoptosis. Clinically, miR-125a expression was linked to increased responsiveness to cisplatin combined with paclitaxel and this resulted in improved outcome. Their data suggests that *miR-125a* may provide a method which allows treatment of resistant cervical cancer. In addition, *miR-125a* may function as a biomarker for predicting resistance to cisplatin and paclitaxel in cervical cancer patients.

Temozolomide, a chemotherapeutic that penetrates the BBB is used for the treatment of the heterogenous glioblastoma and anaplastic astrocytoma. Despite treatment with surgery, chemotherapy, and radiation, survival is maximum 15 months. Hyperactivated STAT3 has been demonstrated to modulate the behavior of gliomas and promote ADR and the STAT3 inhibitor pacritinib in combination with temozolimide has been shown to be effective in glioblastoma overwhelming STAT3/miR-21/PDCD4 signaling [114,115,148]. Moreover, the antipsychotic pimozide, a repurposed STAT3 inhibitor, reduces STAT3, triggers an autophagy-dependent, lysosomal type of cell death and improves survival in GBM cells [149,150]. A rational therapy for the treatment of glioblastoma would be the combination of temozolomide with the STAT3 inhibitor SBT-100, two anticancer compounds that penetrate the BBB [143].

## 5. ADR Development to Targeted Therapies

Sun et al have described numerous studies that have shown that STAT3 activation can result in the failure of many different types of targeted therapies, especially EGFR targeted therapies [20,52]. For example, afatinib-induced STAT3 activation decreases the suppression of lung cancer cells to afatinib, and inhibiting IL-6R/JAK1/STAT3 signaling reverses the resistance. Blocking STAT3 can prevent ADR induced by EGFR inhibitors. Rhein, a lipophilic anthraquinone, sensitizes pancreatic cancer cells to erlotinib by inhibiting STAT3. Alantolactone, a natural sesquiterpene lactone, also sensitizes pancreatic cancer cells to erlotinib and also to afatinib by inhibiting STAT3 [116]. Silibinin, a polyphenolic flavonoid, is a direct inhibitor of STAT3, and it reverses ADR of crizotinib in ALK-rearranged lung cancer cells [68]. In addition, silibinin synergistically improves the response to sorafenib by hepatocellular carcinoma (HCC) cells by blocking STAT3 [151].

Pancreatic adenocarcinoma (PDAC) is a lethal malignancy that presents in late stages and responds poorly to current therapeutic regimens with an overall 5-year survival of 11%. It is characterized by an extensively dense fibrotic tumor stroma with poor vascularity, which hinder the intratumor delivery of anti-neoplastic agents [152]. Examining the response of PDAC to monotherapy with gemcitabine, dasatinib (Src inhibitor), and erlotinib (EGFR inhibitor) reveals that the upregulation of p-STAT3 is causing ADR. When combination therapy of dasatinib and erlotinib is used p-STAT3 is downregulated. This results in tumor collagen (types 1 and 4) and fibrosis to decrease within the tumor stroma in an orthotopic mouse model of PDAC. Here Dosch et al used PANC-1, which has a KRAS (G12D) mutation and constitutive expression of p-STAT3, and BxPC3, which has wild type KRAS and constitutive expression of p-STAT3. Interestingly, the addition of gemcitabine to combine with dasatinib and erlotinib therapy did not reverse the antifibrotic effects of this drug combination [69]. This two-drug combination inhibits the EGFR and Src signaling pathway and reduces p-STAT3. In turn, an increase in tumor vascularity occurs in vivo, and to determine this, treated PDAC were examined for CD31 (PECAM-1) by immunohistochemical (IHC) staining. CD31 is an endothelial marker that is associated with vascular normalization, maturity, and has been correlated with chemotherapeutic response in PDAC [11,28]. Monotherapy with dasatinib or erlotinib versus control showed no significant increase in CD31 positive staining; however, combined in vivo treatment with dasatinib plus erlotinib resulted in significant increase in CD31 positive endothelial cells. This finding was sustained with gemcitabine added to the two-drug combination. Dosch et al showed levels of gemcitabine is nearly undetectable in tumors treated with erlotinib or dasatinib monotherapy or in combination with gemcitabine [69]. When combination therapy with dasatinib plus erlotinib was administered, a marked increase in gemcitabine levels within PDAC tumors was detected. These findings demonstrated that combined Src and EGFR inhibition decreases p-STAT3 activity, which increases the microvascular density within PDAC tumors, which ultimately results in increased delivery of cytotoxic chemotherapy into the tumor mass.

The above orthotopic PDAC studies were conducted on athymic nude mice. Transgenic PKT mice (*Ptf1a*^Cre/+^; LSL-*Kras*^G12D/+^; *Tgfbr2*^flox/flox^) were used for in vivo studies to examine tumor volume with the pancreas and overall survival of combined Src and EGFR inhibition. This immunocompetent, spontaneous mouse model of PDAC underwent treatment with dasatinib, erlotinib, and gemcitabine either alone or in combination. This therapy was continued for 4 weeks, after which the mice were sacrificed and the PDAC tumors harvested for histo-pathology evaluation. In PKT tumors, dasatinib plus erlotinib, and dasatinib plus erlotinib with gemcitabine treatments significantly reduced tumor weight at the end of the study. Furthermore, stromal remodeling of the PDAC tumors occurred as it did in the orthotopic tumors with decreased stromal fibrosis, decreased collagen type 1 and 4, increased microvascular density, and increased number of CD31 positive endothelial cells. Moreover, p-STAT3 levels were significantly decreased with combined treatment of dasatinib plus erlotinib, and dasatinib plus erlotinib with gemcitabine in the PKT tumor samples. These two combination regimens also prevented the progression of PDAC tumors in the PKT mice and increased their overall survival. Furthermore, they have previously shown tumor cell-derived IL1α induces stromal-derived IL-6, reciprocally activating tumor cell-autonomous STAT3 signaling, a well-known indicator of chemoresistance and oncogenic signaling in PDAC [126,133,153].

Lee et al performed extensive and elegant work on defining STAT3 as an escape mechanism for many cancers treated with targeted pharmaceutical therapeutics [154]. They discovered that many drug treated “oncogene addicted” malignancies use a positive feedback loop resulting in STAT3 hyperactivation. As a result, promoting cancer cell proliferation, survival, and decreasing response to targeted drug therapy. This was noted in malignant cells driven by different activated kinases such as HER2, EGFR, MET, ALT, and mutant KRAS [10,13]. MEK inhibition resulted in autocrine activation of STAT3 via FGFR and JAK kinases. Importantly, simultaneous drug suppression of MEK, JAK, and FGFR increased tumor regression. Their data implies that blocking the STAT3 feedback loop enhances the response to a wide range of pharmacologic therapeutics that inhibit pathways of oncogene addiction.

## 6. ADR Development to Monoclonal Antibody Treatment

Monoclonal antibodies (mAbs) to cell surface receptors, [e.g., human epidermal growth factor receptor 2 (HER2)], and to extracellular protein, e.g., vascular endothelial growth factor (VEGF), have greatly improved the treatment of patients with cancer. HER2 is a receptor tyrosine kinase (RTK) that controls differentiation and cell growth signaling pathways. In about 20–25% of breast cancers and in 30% of gastric cancers, HER2 is significantly overexpressed. This results in a very aggressive cancer phenotype and poor prognosis. Trastuzumab is a humanized mAb that binds to an extracellular domain of the HER2 molecule and inhibits its function. It provides significant benefit in patient outcome; however, treatment resistance does develop in some patients. Li et al found that p-STAT3 is hyper-expressed in de novo and acquired trastuzumab-resistant gastric cancer and breast cancer cells [96]. Here, increased STAT3 activation and signaling is caused by elevated levels of IL-6, fibronectin (FN), and EGF in an autocrine manner. This leads to ADR by upregulating the expression of *MUC1* and *MUC4*. Both are downstream targets of p-STAT3. *MUC1* and *MUC4* can induce trastuzumab resistance by maintaining HER2 activation and masking of trastuzumab to prevent HER2 binding, respectively. Knocking down STAT3 expression and blocking STAT3 function with a small molecule inhibitor abrogated STAT3 activation, which allowed trastuzumab sensitivity of resistant cells in vitro and in vivo [89].

Trastuzumab–emtansine (T-DM1) is an antibody drug conjugate made with the trastuzumab mAb linked to a cytotoxic moiety DM1 (a derivative of maytansine) and it was developed to overcome ADR associated with trastuzumab use. T-DM1 has demonstrated great efficacy clinically; however, ADR to its use has emerged and is a significant problem for patients. Wang et al used BT-474/KR cells, a T-DM1 resistant cell line developed from HER2-positive BT-474 breast cancer cells, to show that STAT3 activation mediated by leukemia inhibitory factor receptor (LIFR) overexpression results in T-DM1 resistance. Furthermore, they demonstrated STAT3 inhibition sensitizes resistant cell to T-DM1 both in vitro and in vivo [90].

Anti-VEGF treatments help several types of cancer patients, but ADR can develop with therapy. There are several VEGF pathway inhibitors, which include bevacizumab (anti-VEGF mAb), aflibercept (decoy receptor that binds VEGF-A), and ramucirumab (anti-VEGF receptor 2 mAb), which inhibit tumor growth in preclinical cancer models and improve cancer patients’ survival. Eichten et al developed cell lines from anti-VEGF resistant tumor xenografts and one called A431-V epidermoid carcinoma developed partial resistance to aflibercept [87]. A431-V tumors secreted much more IL-6 and produced higher amounts of p-STAT3 compared to parental tumors. Combined inhibition of IL-6 receptor (IL-6R) and VEGF resulted in enhanced suppression of A431-V tumors. In addition, inhibition of IL-6R increased the suppression of DU145 prostate cancer cells using aflibercept. These DU145s have high endogenous IL-6R activity. These data indicate that ADR to anti-VEGF therapy is mediated in part by increased IL-6/STAT3 signaling in cancer cells. Inhibition of IL-6 signaling on cancer cells can overcome this ADR.

Immune checkpoint inhibitors (ICIs) provide significant benefit to some cancer patients and improve survival in a minority of patients. Moreover, some might even be cured [99]. Tumor-intrinsic resistance is the reason for the lack of response [97]; however, when ICIs are used for the first time, less than 45% respond and most responders eventually develop ADR [101]. ADR has been reported in several types of cancer patients and animal models treated with ICIs due to overactivation of STAT3. This undesirable effect has been observed in the case of anti-PD-1, anti PD-L1, and anti-CTLA-4 antibodies. STAT3 can directly or indirectly regulate these immune checkpoint molecules. There is a clear relation between STAT3 and PD-1, PD-L1, and PD-L2 [6]. STAT3 can increase their expression by direct binding to their promoters [105]. STAT3 binds to the *CD274* (PD-L1) gene promoter and is required for *CD274* gene expression. The NPM/ALK carrying T cell lymphoma (ALK + TCL) cells strongly express PD-L1 that is regulated by STAT3 [100]. These investigators at that time were already suggesting that the treatment of this lymphoma should combine inhibition of both NPM/ALK and STAT3. Under hypoxic conditions overactivated STAT3 interacts with PD-L1 and enables its nuclear translocation by the importin α and β pathways [95]. It was shown that in nasopharyngeal carcinoma, LMP1 upregulates PD-L1 through STAT3, AP-1, and NF-kB [94]. In gastric cancer, the suppressor gene *miR-502-5p* reduces PD-L1 expression through inhibition of the CD40/STAT3 pathway [106].

To obtain better results, combinations must be used. Nivolumab plus ipilimumab improved outcomes in 43% of patients with metastatic renal cell carcinoma but ADR was found in the rest of patients receiving this combination. Nonresponders exhibited significant increases in cytokines and higher levels of p-STAT3. The addition of a STAT3 inhibitor to the combination of the two ICIs showed significant tumor growth inhibition in a syngeneic model [92]. These investigators suggest that anti-PD-1 therapy administered along with a STAT3 inhibitor is a rational combination and should be further explored. In patients with melanoma, less than 20% respond to ICIs. Studies in a melanoma mouse model showed that the addition of STAT3 inhibitors to an ICI increases the response to the ICI-resistant tumor. These data suggest that the combination of ICIs with STAT3 inhibitors might be effective in patients with melanoma [96]. In drug-resistant BRAF-mutant melanoma, a combined blockade of STAT3 and PD-1 overcomes resistance [8,14,19,98]. In PDAC, the addition of MEK inhibitors plus STAT3 inhibitors to Nivolumab overcomes ICI resistance [93].

Ipilimumab is efficacious only in a subset of patients with prostate cancer. In a syngeneic prostate cancer mouse model, the combination of an anti-CTLA-4 with a STAT3 inhibitor significantly inhibited tumor growth and enhanced survival possibly by blocking STAT3 mediated resistance mechanisms such as Tregs in the immunosuppressive environment. These investigators raise the possibility that STAT3 inhibition in combination with anti-CTLA-4 could constitute a future novel treatment approach in advanced prostate cancer [103]. In summary, it appears that combining a STAT3 inhibitor with an ICI is an attractive way to prevent the development of ADR and increase their efficacy [92,93,96,106].

## 7. Discussion

Several mechanisms involved in the development of ADR have been studied in animal models of several types of cancer and in patients. As result of the unique cytoplasmatic location of the STAT3 signaling pathway downstream of major pathways involved in cancer and the significant crosstalk that occurs among them, it is activated by the inhibition of many other pathways and in most types of cancers becomes the constitutively activated p-STAT3. This explains why the administration of any anticancer therapeutic and the inhibition of its specific pathway through the crosstalk relations induces the production of p-STAT3 that appears to be responsible for the various changes that end in the development of ADR and the loss of their therapeutic effects. p-STAT3 alters autophagy and anoikis, two apoptotic processes that normally eliminates unwanted cells and the lack or reduction of their actions contribute to ADR and the progression of the tumor or hematological cancer. The physiological relation between STAT3, and p53, or NNMT is altered when STAT3 becomes constitutively activated. These alterations act as factors contributing to the development of ADR. Participation of STAT3 in ARD after administration of any type of anticancer therapy—including the newer targeted agents such as the eight new ICIs and the two KRAS inhibitors—indicates that the use of a STAT3 inhibitor should be part of any rational pharmacological treatment including radiotherapy. The inclusion of a STAT3 inhibitor in anticancer regimens increases their efficacy and most likely prevents the development of ADR. Some natural products, such as curcumin, are STAT3 inhibitors. A limitation of curcumin is the low oral bioavailability; however, new delivery technologies have improved it oral absorption [155]. When ADR is already present, the administration of a STAT3 inhibitor reverses it and restores the efficacy of the anticancer therapeutic.

Many investigators have shown that STAT3 inhibition can reverse ADR, restore the efficacy of anticancer agents [118,156], enhance anti-cancer immune responses, and rescue the suppressed immunologic system [157]. The treatment of cancer usually requires combination therapy, and two or more agents might be needed. As soon as direct STAT3 inhibitors reach the market, a rational combination for the pharmacologic treatment of cancer patients should include a STAT3 inhibitor to prevent and or reverse ADR and thereby increase the efficacy and duration of the therapeutic regimen.

## Figures and Tables

**Figure 1 ijms-24-04722-f001:**
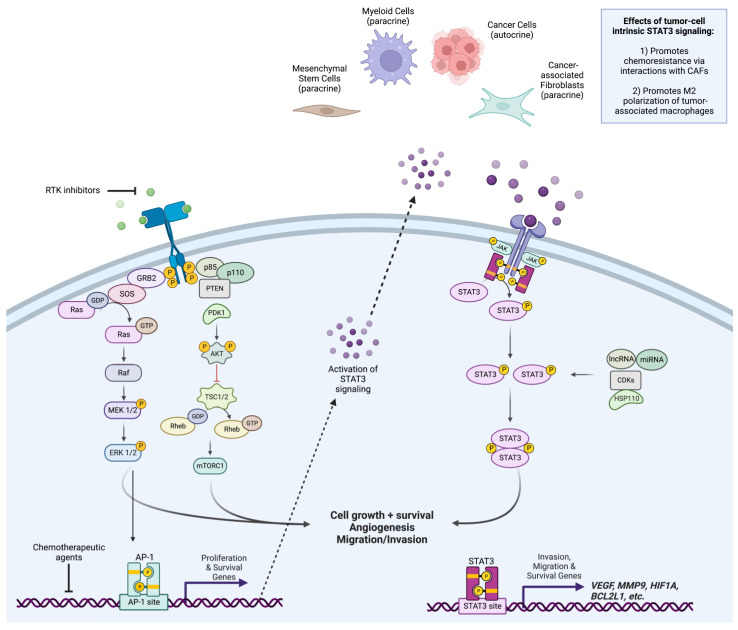
Graphical schematic depicting molecular mechanism of STAT3-mediated acquired drug resistance in response to receptor tyrosine kinases within cancer cells. Adapted from Yang et al. [144].

**Table 1 ijms-24-04722-t001:** Chemical structures and functions for standard chemotherapies, receptor tyrosine kinases, hormonal therapies, and monoclonal antibodies utilized to treat various cancers.

Generic Name (Brand Name)	Chemical Structure & Formula	Drug Target	Mechanism of Action	Reference Numbers
Cisplatin (Platinol)	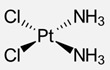 cis-[Pt(NH_3_)_2_Cl_2_]	The N7 reactive center on purine nucleotide residues of DNA	Interferes with DNA replication, commonly by forming 1,2 intra- or interstrand crosslinks	[34,45,46,47,48,49,50,51,52,53,54,55]
Docetaxel (Taxotere)	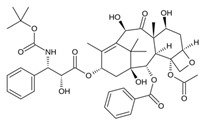 C_43_H_53_NO_14_	Intracellular microtubular network and Bcl-2	Stabilizes microtubule structures to impair depolymerization and attenuation of Bcl-2 expression effects	[30,47]
Doxorubicin (Adriamycin)	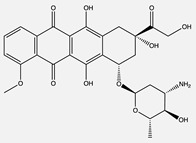 C_27_H_29_NO_11_	DNA and the topoisomerase II complex of DNA	Intercalates DNA and inhibits progression of topoisomerase II to stop the replication process	[30,31,32,33,107,108,109,110,111]
Gemcitabine (Gemzar)	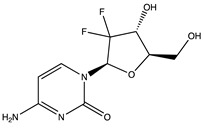 C_9_H_11_F_2_N_3_O_4_	DNA nucleotides	Acts as a cytidine analog to replace nucleotides during DNA replication, causing cell arrest and apoptosis	[34,35,36,37,38,39,40,41,42,43]
Paclitaxel (Taxol and Others)	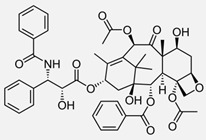 C_47_H_51_NO_14_	Tublin subunits of intracellular microtubular network	Hyperstabilizes microtubules to impair disassembly and ultimately block mitosis progression	[60,61,62,63,64,112]
Cyclophosphamide (Cytoxan)	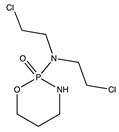 C_7_H_15_Cl_2_N_2_O_2_P	DNA strands at guanine N-7 positions	Forms intrastrand and interstrand DNA cross-linkages	[9]
Temozolomide (Temodar)	C_6_H_6_N_6_O_2_	The N7 position of guanine, N3 of adenine and the O6 position of guanine	Broken down to a methyl 80-diazonium cation that methylates adenosine and guanine residues, causing DNA lesions to trigger apoptosis	[55,56,57,58,59,113,114,115]
Afatinib (Gilotrif)	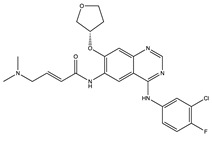 C_24_H_25_ClFN_5_O_3_	ErbB family: EGFR (ErB1), HER2 (ErbB2), and HER4 (ErbB4)	Covalently binds to kinase domains, irreversibly preventing autophosphorylation by homo- or heterodimers	[36,59,65,66]
Bortezomib (Velcade)	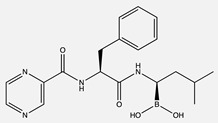 C_19_H_25_BN_4_O_4_	26S proteasome	Inhibits a ubiquitin-proteasome, preventing the degradation of pro-apoptotic factors to induce cell cycle arrest	[12,16,21,24,25,27]
Crizotinib (Xalkori)	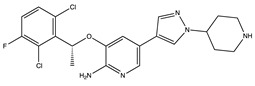 C_21_H_22_Cl_2_FN_5_O	ALK, HGFR, cMET, cROS, and RON	Inhibits phosphorylation in and creates an inactive protein confirmation	[67]
Dasatinib (Sprycel)	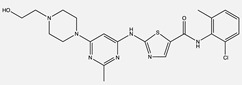 C_22_H_26_ClN_7_O_2_S	BCR-ABL, SRC family, c-KIT, EPHA2, and PDGFRβ	Competitively binds to ATP binding site of the kinase domain	[68]
Erlotinib (Tarceva)	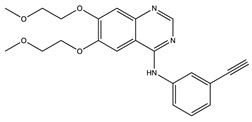 C_22_H_23_N_3_O_4_	Epidermal Growth Factor Receptor (EGFR)	Inhibits intracellular phosphorylation (not fully characterized)	[69,70,71,72,73]
Gefitinib (Iressa)	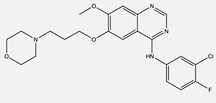 C_22_H_24_ClFN_4_O_3_	EGFR	Competitively binds to ATP binding site of the kinase domain	[5,7,15,23,116]
Lapatinib (Tykerb)	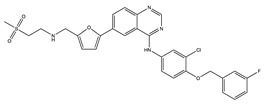 C_29_H_26_ClFN_4_O_4_S	EGFR and Epidermal Growth Factor Receptor 2 (HER2)	Competitively binds to intracellular ATP binding site of the receptor	[10,13]
Vemurafenib (Zelboraf)	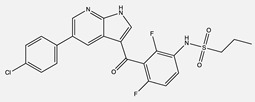 C_23_H_18_ClF_2_N_3_O_3_S	BRAF (V600E)	Competitively binds to kinase domain of mutant form of BRAF	[8,11,14,19,97]
Imatinib (Gleevec)	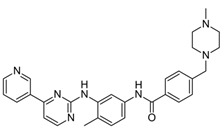 C_29_H_31_N_7_O	BCR-ABL, ABL, CSF1R, FLT-3, c-KIT, PDGFR	Competitively binds to activate site of kinase to block phosphorylation	[17,18,26]
Enzalutamide (Xtandi)	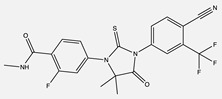 C_21_H_16_F_4_N_4_O_2_S	Androgen Receptor (AR)	Competitively binds to the androgen receptor, preventing gene expression of AR targets	[66,67,68]
Flutamide (Eulexin)	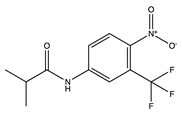 C_11_H_11_F_3_N_2_O_3_	Androgen Receptor	Competitively binds to the androgen receptor blocking binding and uptake of testosterones	[74,75,76]
Tamoxifen (Nolvadex)	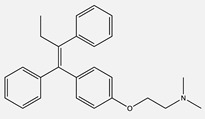 C_26_H_29_NO	Estrogen Receptor (ER)	Competitively inhibits binding of estrogen to its receptor	[79,80,81,82]
Cetuximab (Erbitux)	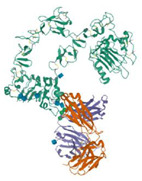 C_6484_H_10042_N_1732_O_2023_S_36_	EGFR	Competitively inhibits the binding of epidermal growth factor (EGF) and other ligands to block the extended conformation of EGFR	[83,84,85]
Trastuzumab (Herceptin)	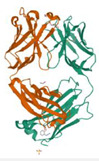 C_6470_H_10012_N_1726_O_2013_S_42_	HER2	Binds to extracellular ligand-binding domain, blocking its cleavage and inducing downregulation of the receptor	[87,88,89,90,96]
Bevacizumab (Avastin)	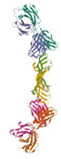 C_6538_H_10034_N_1716_O_2033_S_44_	Vascular endothelial growth factor (VEGF)	Binds and inactivates circulating VEGF, preventing its binding to receptor	[86]
Pembrolizumab (Keytruda)	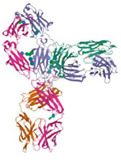 C_6504_H_10004_N_1716_O_2036_S_46_	PD-1 (CD279)	Binds to PD-1 on T cells to antagonize interactions with ligands, PD-L1 and PD-L2	[100]
Nivolumab (Opdivo)	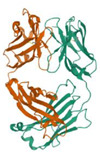 C_6362_H_9862_N_1712_O_1995_S_42_	PD-1	Binds to PD-1 on T cells to antagonize interactions with ligands, PD-L1 and PD-L2	[91,92,101,103]
Ipilimumab (Yervoy)	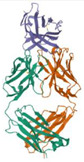 C_6572_H_10126_N_1734_O_2080_S_40_	CTLA-4 (CD152)	Binds CTLA-4, preventing its binding to CD28 on antigen presenting cells or T cells	[91,92,95,102,105]
Atezolizumab (Tecentriq)	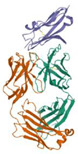 C_6446_H_9902_N_1706_O_1998_S_42_	PD-L1 (CD274)	Selectively binds to PD-L1, blocking its interaction with PD-1 or B7-1	[6]

## Data Availability

Not applicable.

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
