# Peer review of "Overcoming Acquired Drug Resistance to Cancer Therapies through Targeted STAT3 Inhibition"

_ijms, 2023, doi:10.3390/ijms24054722_

Round 1

Reviewer 1 Report

The manuscript “Overcoming acquired drug resistance to cancer therapies through targeted STAT3 inhibition” by Sunanda Singh and co-authors to evaluate scientific and medical literature which elucidate STAT3-mediated mechanisms of resistance to cancer therapeutics. Herein, we have found that at least 24 different anti-neoplastic agents standard toxic chemotherapeutic agents, targeted kinase inhibitors, anti-hormonal agents, and monoclonal antibodies that utilize the STAT3 signaling pathway as one mechanism of developing therapeutic resistance. Targeting STAT3, in combination with existing anti-neoplastic agents, may prove to be a successful therapeutic strategy to either prevent or even overcome ADR to standard and novel cancer therapies. However, there are concerns that must be taken into account before the work can be reconsidered for publication.

1.      Table 1: Please list cancer type.

2. Figure 1: Any interaction between cancer cell and CAF or cancer cell and myeloid or cancer cell and mesenchymal?

Author Response

Our responses in red.

The manuscript “Overcoming acquired drug resistance to cancer therapies through targeted STAT3 inhibition” by Sunanda Singh and co-authors to evaluate scientific and medical literature which elucidate STAT3-mediated mechanisms of resistance to cancer therapeutics. Herein, we have found that at least 24 different anti-neoplastic agents standard toxic chemotherapeutic agents, targeted kinase inhibitors, anti-hormonal agents, and monoclonal antibodies that utilize the STAT3 signaling pathway as one mechanism of developing therapeutic resistance. Targeting STAT3, in combination with existing anti-neoplastic agents, may prove to be a successful therapeutic strategy to either prevent or even overcome ADR to standard and novel cancer therapies. However, there are concerns that must be taken into account before the work can be reconsidered for publication.

  1. Table 1: Please list cancer type. We are not sure where in the paper the reviewer recommends we list the cancers. Table 1 has a very large list of anti-neoplastic agents which collectively are used for almost every cancer know.
  2. Figure 1: Any interaction between cancer cell and CAF or cancer cell and myeloid or cancer cell and mesenchymal? Yes, there is interactions between cancer cells and all these stromal cell types. We will mention this.

Reviewer 2 Report

In this manuscript authors reviewed the current literature regarding the role of STAT3 inhibition in prevent or even overcome acquired drug resistance (ADR).

I greatly appreciate the work performed by authors since the manuscript is very interesting and generally well written and easy to read. However, some points deserve to be improved. In particular:

1.    Lines 120-122: It deserves to be specified that autophagy is also involved in cancer chemoresistance; please revise the manuscript providing details on this topic.

2.    Paragraph 4: authors effectively underline the works where STAT3 was demonstrated to promote acquiring drug resistance in several cancers. Nonetheless, the exact mechanism seems to be still elusive. One possibility might be linked to the enzyme nicotinamide N-methyltransferase (NNMT). NNMT has been reported to be upregulated in many tumors (PMID: 34439880; PMID: 34827592), and some studies demonstrated that it can be regulated by STAT3 expression (PMID: 17922140) while in other studies seems to be that NNMT regulates STAT3 expression (PMID: 35851575). Since NNMT inhibition sensitizes cells to chemotherapy (PMID: 33907844; PMID: 31101119), it plays a crucial role in developing chemoresistance. Thus, this enzyme, which is under the control of STAT3 or may be an inducer of STAT3, is strictly connected to STAT3 role in developing drug resistance. Several NNMT inhibitors have been developed for cancer treatment and could be tested for their effect on STAT3 expression (PMID: 34704059; PMID: 34424711; PMID: 34572571). All these considerations should be included in the manuscript since they would enrich the discussion and open new perspectives.

3.    Please improve figure 1 specifying the name of several crucial genes under the control of Stat3 involved in invasion, migration and survival.

4.    Discussionin this section authors should also mention the presence of natural inhibitors of JAK/STAT3 signalling. For instance, curcumin is one of the most efficient showing a strong inhibition of STAT3 activation. This is an important point to highlight since curcumin may be used in cotreatment with the chemotherapeutics discussed in this review reducing or avoiding chemotherapy resistance onset.

5.    Genes must be written in Italic

6.    A summary table resuming the studies discussed in each section should be added

7.    An accurate revision of punctuation is recommended

Author Response

Our responses in red.

In this manuscript authors reviewed the current literature regarding the role of STAT3 inhibition in prevent or even overcome acquired drug resistance (ADR).

I greatly appreciate the work performed by authors since the manuscript is very interesting and generally well written and easy to read. However, some points deserve to be improved. In particular:

  1. Lines 120-122: It deserves to be specified that autophagy is also involved in cancer chemoresistance; please revise the manuscript providing details on this topic. We agree and will make this change.

  1. Paragraph 4: authors effectively underline the works where STAT3 was demonstrated to promote acquiring drug resistance in several cancers. Nonetheless, the exact mechanism seems to be still elusive. One possibility might be linked to the enzyme nicotinamide N-methyltransferase (NNMT). NNMT has been reported to be upregulated in many tumors (PMID: 34439880; PMID: 34827592), and some studies demonstrated that it can be regulated by STAT3 expression (PMID: 17922140) while in other studies seems to be that NNMT regulates STAT3 expression (PMID: 35851575). Since NNMT inhibition sensitizes cells to chemotherapy (PMID: 33907844; PMID: 31101119), it plays a crucial role in developing chemoresistance. Thus, this enzyme, which is under the control of STAT3 or may be an inducer of STAT3, is strictly connected to STAT3 role in developing drug resistance. Several NNMT inhibitors have been developed for cancer treatment and could be tested for their effect on STAT3 expression (PMID: 34704059; PMID: 34424711; PMID: 34572571). All these considerations should be included in the manuscript since they would enrich the discussion and open new perspectives. We agree and will include this topic in the paper.

  1. Please improve figure 1 specifying the name of several crucial genes under the control of Stat3 involved in invasion, migration and survival. We will do this.

  1. Discussionin this section authors should also mention the presence of natural inhibitors of JAK/STAT3 signalling. For instance, curcumin is one of the most efficient showing a strong inhibition of STAT3 activation. This is an important point to highlight since curcumin may be used in cotreatment with the chemotherapeutics discussed in this review reducing or avoiding chemotherapy resistance onset. We agree and will include curcumin.

  1. Genes must be written in Italic. Agreed.

  1. A summary table resuming the studies discussed in each section should be added. We are not sure what is meant by “resuming” in your sentence.

  1. An accurate revision of punctuation is recommended. Agree.

Reviewer 3 Report

In their manuscript, Singh et al. reviewed the mechanisms of drug resistance involving STAT3 signaling. The authors describe a plethora of studies suggesting that ADR following several types of therapies correlates with an increase in p-STAT3.

The review is very well written, clearly organized and easy to read. However, the authors should answer the following comments to improve the quality of their manuscript:

- A figure summarizing the signaling pathways involving STATs (STAT chapter of the review) would help non expert readers.  

- A table summarizing the combinations of treatments with STAT3 that have proven to delay or revert ADR and in which cancer would also be a great addition.

- When describing pathways or mechanisms of action, authors should consider using the present tense instead of the past tense. They should also make sure that the same tense is used throughout any given paragraph.

- Line 199-201: the idea of a combination of both effects on normal cells and activation of STAT3 is repeated twice.

- Line 292: The beginning of the paragraph is related to the previous paragraph and should not be disconnected from it.

Line 311-313 and 327-329: the two sentences should be only one sentence in both situations.

Author Response

Our responses are in red.

In their manuscript, Singh et al. reviewed the mechanisms of drug resistance involving STAT3 signaling. The authors describe a plethora of studies suggesting that ADR following several types of therapies correlates with an increase in p-STAT3.

The review is very well written, clearly organized and easy to read. However, the authors should answer the following comments to improve the quality of their manuscript:

- A figure summarizing the signaling pathways involving STATs (STAT chapter of the review) would help non expert readers.  This is shown on the right side of Figure 1.

- A table summarizing the combinations of treatments with STAT3 that have proven to delay or revert ADR and in which cancer would also be a great addition. We are not aware of any commercially available drug that inhibits STAT3 and has proven to delay ADR. Metformin inhibits STAT3, and it has been shown in retrospective studies that patients taking metformin have a reduced incidence of breast cancer. Despite this association, no prospective clinical trial has successfully shown significant benefit with the use of metformin in cancer patients. Celecoxib also inhibits STAT3, and separately has been shown to reduce the incidence of colon cancer. Given this, celecoxib is not used in the treatment of any malignancies that we are aware of.

- When describing pathways or mechanisms of action, authors should consider using the present tense instead of the past tense. They should also make sure that the same tense is used throughout any given paragraph. We will make this change.

- Line 199-201: the idea of a combination of both effects on normal cells and activation of STAT3 is repeated twice. Change made.

- Line 292: The beginning of the paragraph is related to the previous paragraph and should not be disconnected from it. Change made.

Line 311-313 and 327-329: the two sentences should be only one sentence in both situations. These sentences are in two separate paragraphs and are unrelated. Please recheck.

Round 2

Reviewer 1 Report

The revised manuscript “Overcoming acquired drug resistance to cancer therapies through targeted STAT3 inhibition” have adequately addressed my previous concerns and the paper is now acceptable for publication.

Reviewer 2 Report

Manuscript has been improved and can be published.